# Video Anomaly Detection via Semantic Attributes

## Abstract

Video anomaly detection (VAD) is a challenging computer vision task with many practical applications. As anomalies are inherently ambiguous, it is essential for users to understand the reasoning behind a system's decision in order to determine if the rationale is sound. In this paper, we propose a simple but highly effective method that pushes the boundaries of VAD accuracy and interpretability using attribute-based representations. Our method represents every object by its velocity and pose. The anomaly scores are computed using a density-based approach. Surprisingly, we find that this simple representation is sufficient to achieve state-of-the-art performance in ShanghaiTech, the largest and most complex VAD dataset. Combining our interpretable attribute-based representations with implicit, deep representation yields state-of-the-art performance with a $99.1\%, 93.6\%$, and $85.9\%$ AUROC on Ped2, Avenue, and ShanghaiTech, respectively. Our method is accurate, interpretable, and easy to implement.

## 1 Introduction

Video anomaly detection (VAD) is a key goal of video surveillance but is very challenging. One of the most common VAD settings is the one-class classification (OCC). In this setting, only normal videos are seen during the training stage without any anomalies. At deployment, the trained model is required to distinguish between normal events and those that are abnormal in *a semantically meaningful way*. The key difficulty is that the difference between patterns that are semantically meaningful and those that are not is subjective. In fact, two human operators may disagree on whether an event is anomalous. Furthermore, as no labeled anomalies are provided for training, it is not possible to directly learn the discriminative patterns.

VAD has been researched for decades, but the advent of deep learning has brought significant breakthroughs. Recent approaches to anomaly detection follow two main directions: (i) handcrafted priors for self-supervised learning: many methods designed auxiliary tasks such as rotation prediction, invariance to handcrafted augmentations, and predicting the arrow of time and rate of temporal flow. These approaches dominate VAD. (ii) Representation extraction using pretrained encoder: a two-stage approach which first computes representations using pretrained encoders (such as ResNet pretrained on ImageNet), followed by standard density estimation such as $k$NN or Mahalanobis distance. This approach is successful in image anomaly detection and segmentation. The issue with both approaches is that the representations that they learn are opaque and non-interpretable. As anomalies are ambiguous, it is essential that the reasoning is made explicit so that a human operator could understand if the criteria for the decision are justified.

Most state-of-the-art anomaly detection methods are not interpretable, despite their use in safety-critical applications. In this paper, we follow a new direction: representing data using semantic attributes which are meaningful to humans and therefore easier to interpret. Our method extracts representations consisting of the velocity and pose attributes, which were found to be important in previous work (Markovitz et al., 2020; Georgescu et al., 2021a). We use these representations to score anomalies by density estimation. Our method classifies frames as anomalous if their velocity and/or pose take an unusual value. This allows automatic interpretation; the attribute taking an unusual value is interpreted to be the rationale behind the decision (see Fig. 1).

It is surprising that our simple velocity and pose representations achieves state-of-the-art performance on the largest and most complex VAD dataset, with 85.9% AUROC in ShanghaiTech. While

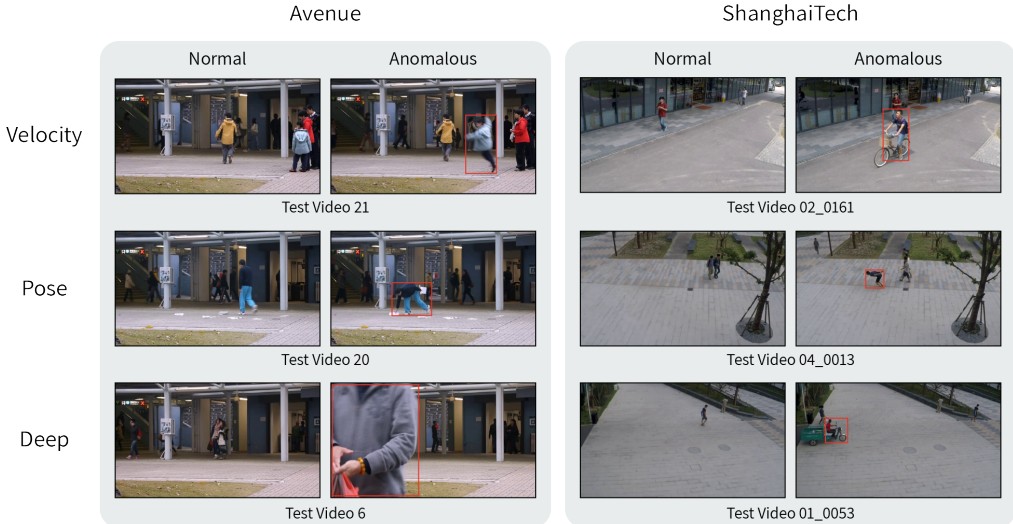

Figure 1: Human-interpretable visualizations on Avenue and ShanghaiTech. We present the most normal and anomalous frames for each feature. For anomalous frames, we visualize the bounding box of the object with the highest anomaly score. Best viewed in color.

our attribute-based representation is very powerful, there are concepts that are not adequately represented by it. The reason is that some attributes cannot simply be quantified using semantic human attributes. Consequently, to model the residual attributes, we couple our explicit attribute-based representation with an implicit, deep representation, obtaining the best of both worlds. Our final method achieves state-of-the-art performance on the three most commonly reported datasets while being highly interpretable. The advantages of our method are three-fold:

1. Achieving state-of-the-art results in the three most commonly used public datasets: 99.1%, 93.6%, 85.9% AUROC on Ped2, Avenue and ShanghaiTech.
2. Making interpretable decisions, important in critical environments where human understanding is key.
3. Being easy to implement.

## 2 RELATED WORK

Classical video anomaly detection methods were typically composed of two steps: handcrafted feature extraction and anomaly scoring. Some of the manual features that were extracted were: optical flow histograms (Chaudhry et al., 2009; Colque et al., 2016) and SIFT (Lowe, 2004). Commonly used scoring methods include: density estimation (Eskin et al., 2002; Glodek et al., 2013; Latecki et al., 2007), reconstruction (Jolliffe, 2011), and one-class classification (Scholkopf et al., 2000).

In recent years, deep learning has gained in popularity as an alternative to these early works. The majority of video anomaly detection methods utilize at least one of three paradigms: reconstruction-based, prediction-based, skeletal-based, or auxiliary classification-based methods.

**Reconstruction & prediction based methods.** In the reconstruction paradigm, the normal training data is typically characterized by an autoencoder, which is then used to reconstruct input video clips. The assumption is that a model trained solely on normal training clips will not be able to reconstruct anomalous frames. This assumption does not always hold true, as neural networks can often generalize to some extent out-of-distribution. Notable works are (Nguyen & Meunier, 2019; Chang et al., 2020; Hasan et al., 2016; Luo et al., 2017b; Yu et al., 2020; Park et al., 2020).

Prediction-based methods learn to predict frames or flow maps in video clips, including inpainting intermediate frames, predicting future frames, and predicting human trajectories (Feng et al., 2021b; Chen et al., 2020; Lu et al., 2019; Wang et al., 2021; Feng et al., 2021a; Yu et al., 2020). Additionally, some works take a hybrid approach combining the two paradigms (Liu et al., 2021b; Zhao et al.,

2017; Ye et al., 2019; Tang et al., 2020). As these methods are trained to optimize both objectives, input frames with large reconstruction or prediction errors are considered anomalous.

**Self-supervised auxiliary tasks.** There has been a great deal of research on learning from unlabeled data. A common approach is to train neural networks on suitably designed auxiliary tasks with automatically generated labels. Tasks include: video frame prediction (Mathieu et al., 2016), image colorization (Zhang et al., 2016; Larsson et al., 2016), puzzle solving (Noroozi & Favaro, 2016), rotation prediction (Gidaris et al., 2018), arrow of time (Wei et al., 2018), predicting playback velocity (Doersch et al., 2015), and verifying frame order (Misra et al., 2016). Many video anomaly detection methods use self-supervised learning. In fact, self-supervised learning is a key component in the majority of reconstruction-based and prediction-based methods. SSMTL (Georgescu et al., 2021a) trains a CNN jointly on three auxiliary tasks: arrow of time, motion irregularity, and middle-box prediction, in addition to knowledge distillation. Jigsaw-Puzzle (Wang et al., 2022) trains neural networks to solve spatio-temporal jigsaw puzzles. The networks are then used for VAD.

**Skeletal methods.** Such methods rely on a pose tracker to extract the skeleton trajectories of each person in the video. Anomalies are then detected using the skeleton trajectory data. Our attribute-based method outperforms previous skeletal methods (e.g., Markovitz et al. (2020); Rodrigues et al. (2020); Yu et al. (2021); Sun & Gong (2023)) by a large margin. Different from skeletal approaches, our method does not require pose tracking, which is extremely challenging in crowded scenes. Our pose features only use a single frame, while our velocity features only require a pair of frames. In contrast, skeletal approaches require pose tracking across many frames, which is expensive and error-prone. It is also important to note that skeletal features by themselves are ineffective in detecting non-human anomalies, therefore, being insufficient for providing a complete VAD solution.

**Object-level video anomaly detection.** Early methods, both classical and deep learning, operated on entire video frames. This proved difficult for VAD as frames contain many variations, as well as a large number of objects. More recent methods (Georgescu et al., 2021a; Liu et al., 2021b; Wang et al., 2022) operate at the object level by first extracting object bounding boxes using off-the-shelf object detectors. Then, they detect if each object is anomalous. This is an easier task, as objects contain much less variation than whole frames. Object-based methods yield significantly better results than frame-level methods.

It is often believed that due to the complexity of realistic scenes and the variety of behaviors, it is difficult to craft features that will discriminate between them. As object detection was inaccurate prior to deep learning, classical methods were previously applied at the frame level rather than at the object level, and therefore underperformed on standard benchmarks. We break this misconception and demonstrates that it is possible to craft semantic features that are both accurate and interpretable.

## 3 PRELIMINARIES

In the VAD task, we are given a training set $\{c_1, c_2...c_{N_c}\} \in \mathcal{X}_{train}$ consisting of $N_c$ video clips that are all normal (i.e., do not contain any anomalies). Each clip $c_i$ is comprised of $N_i$ frames, $c_i = [f_{i,1}, f_{i,2}, ...f_{i,N_i}]$. Given an inference clip $c$ the goal is to classify each frame $f \in c$ as being normal or anomalous. Each frame $f$ is represented using a function $\phi(f) \in \mathbb{R}^d$, where $d \in \mathbb{N}$ is the feature dimension. Next, an anomaly scoring function $s(\phi(f))$ computes the anomaly score for frame $f$. The frame is classified as anomalous if $s(\phi(f))$ exceeds a constant threshold.

## 4 METHODOLOGY

### 4.1 OVERVIEW

We compute an anomaly score based on density estimation of object-level feature descriptors. This is done in three stages: pre-processing, feature extraction, and density estimation. In the pre-processing stage (i) an off-the-shelf motion estimator is applied to predict the optical flow of each frame; (ii) an off-the-shelf object detector is used to localize and classify the bounding boxes of all objects within a frame. The outputs of both models are used to extract object-level velocity, pose, and deep representations (see Sec. 4.3). Finally, the anomaly score of each test frame is computed using density estimation. The computation of object-level features is illustrated in Fig. 2.

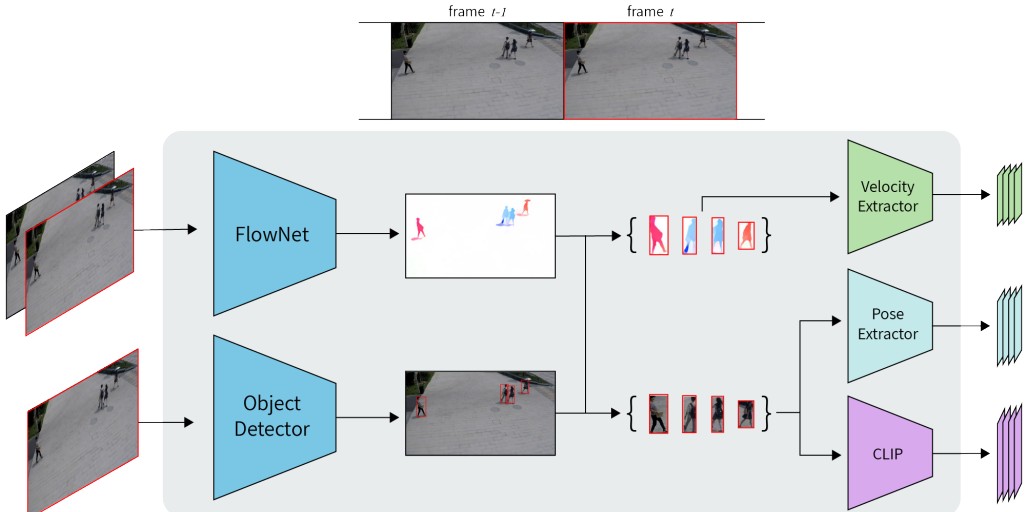

Figure 2: An overview of our proposed method for extracting explicit attribute-based representations, and implicit deep representations. As a first step, we extract optical flow maps and bounding boxes for all of the objects in the frame. We then crop each object from the original image and its corresponding flow map. Our representation consists of velocity, pose, and deep (CLIP) features.

## 4.2 PRE-PROCESSING

Anomalous objects in video clips typically exhibit unusual motions or activities. Therefore, we rely on representations that are linked to objects and motions.

**Optical flow.** Our method uses optical flow as a preliminary stage for inferring object movement. It is computed between every pair of two successive frames. We extract the optical flow map, denoted by $o$ for each frame $f \in c$ in every video clip $c$ using an off-the-shelf optical flow model.

**Object detection.** Our method models frames by representing every object individually. This follows many recent papers, e.g., (Georgescu et al., 2021a; Liu et al., 2021b; Wang et al., 2022) that found object-based representations to be more effective than global, frame-level representations. Similarly to the recent papers, we first detect all objects in each frame using an off-the-shelf object detector. Formally, our object detection generates a set of $m$ bounding boxes $b_1, b_2...b_m$ for each frame, with corresponding class labels $y_1, y_2, ..., y_m$.

## 4.3 FEATURE EXTRACTION

Our method represents each object by two attributes: velocity and pose.

**Velocity features.** Our working hypothesis is that unusual velocity is a relevant attribute for identifying anomalies in video. As objects can move in both $x$ and $y$ axes and both the magnitude (speed) and orientation of the velocity may be anomalous, we compute velocity features for each object in each frame. We begin by cropping the frame-level optical flow map by the bounding box of each object as detected by the object detector. Following this step, we obtain a set of cropped object flow maps, as illustrated in Fig. 2. These flow maps are then rescaled to a fixed size of $H_{flow} \times W_{flow}$. Next, we represent each flow map with the average motion for each orientation, where orientations are quantized into $B \in \mathbb{N}$ equi-spaced bins (a similar idea as Chaudhry et al. (2009)). The final representation is a $B$-dimensional vector consists of the average flow magnitudes of the flow vectors in each bin, as illustrated in Fig. 3. This representation is capable of describing motion in both the radial and tangential directions. We denote our velocity feature extractor as: $\phi_{velocity} : H_{flow} \times W_{flow} \to \mathbb{R}^B$.

**Pose features.** Irregular human activity is often anomalous. While a full understanding of activity requires temporal features, we find that human pose from even a single frame may provide a sufficiently discriminative signal of irregular activities. We represent human pose by its body landmark

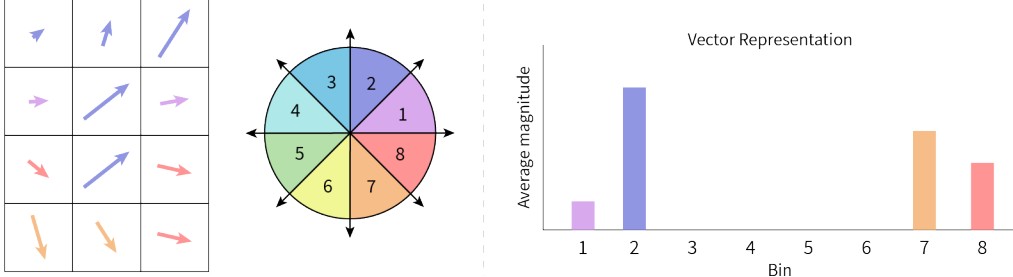

Figure 3: An illustration of our velocity feature vector $\phi_{velocity}$. *Left:* Orientations are quantized into $B = 8$ equi-spaced bins, and each optical flow vector in the object's bounding box is assigned to one-directional bin. *Right:* The average magnitudes of the optical flow vectors in each bin give a velocity feature vector of dimension $B$. Best viewed in color.

positions. Our method obtains pose feature descriptors for each human object $o$ using an off-the-shelf keypoints extractor, denoted by $\hat{\phi}_{pose}(o) \in \mathbb{R}^{2 \times d}$, where $d \in \mathbb{N}$ is the number of keypoints. In practice, we used AlphaPose (Fang et al., 2017), which we found to work well. The output of the keypoints extractor is the pixel coordinates of each landmark position. We perform a simple normalization stage to ensure that the keypoints are invariant to the position and size of the human. We first subtract from each landmark, the coordinates of the top-left corner of the object bounding box. We then scale the $x$ and $y$ axes so that the object bounding box has a final size of $H_{pose} \times W_{pose}$ (where $H_{pose}, W_{pose}$ are constants). Formally, let $l \in \mathbb{R}^2$ be the top-left corner of the human bounding box. The pose description becomes:

$$\phi_{pose}(o) = \begin{pmatrix} \frac{H_{pose}}{height(o)} & 0 \\ 0 & \frac{W_{pose}}{width(o)} \end{pmatrix} (\hat{\phi}_{pose}(o) - l) \tag{1}$$

Where $height(o), width(o)$ is the object $o$ bounding box height and width respectively. Finally, we flatten $\phi_{pose}$ to obtain the final pose feature vector.

**Deep features.** While our attribute-based representation is already very powerful, it is sometimes insufficiently expressive to detect all anomalies. Powerful deep features are very expressive, bundling together many different attributes. Hence, we use implicit, deep representations to model the residual attributes which are not described by velocity and pose. In image anomaly detection, implicit representations are pretrained on external, generic datasets and then transferred to the anomaly detection task. Previous work (Reiss et al., 2021; Reiss & Hoshen, 2023) showed that coupling such powerful representations with simple anomaly detection classifiers (e.g., $k$NN) can achieve outstanding results. Concretely, our implicit representation is computed using a pretrained CLIP encoder (Radford et al., 2021), denoted by $\phi_{deep}(.)$, to represent the bounding box of each object in each frame.

## 4.4 DENSITY ESTIMATION

We use density estimation for scoring samples as normal or anomalous, where low estimated density is indicative of anomaly. To estimate the density, we fit a separate estimator for each feature. For velocity features, which are lower dimensional, we use a GMM estimator. As our pose and deep features are high-dimensional and are not assumed to obey particular parametric assumptions, we estimate their density using $k$NN. I.e., we compute the $L_2$ distance between feature $x$ of a target object and the $k$ exemplars in the corresponding training feature set. A comparison of different exemplar selection methods is in Sec. 5.4. We denote our density estimators by $s_{velocity}(.), s_{pose}(.), s_{deep}(.)$.

**Score calibration.** Combining the three density estimators requires calibration. To do so, we estimate the distribution of anomaly scores on the normal training set. We then scale the scores using min-max normalization. The $k$NN used for scoring pose and deep features present a subtle point. When computing $k$NN on the training set, the exemplars must not be taken from the same clip as the target object. The reason is that the same object appears in nearby frames with virtually no variation, distorting $k$NN estimates. Instead, we compute the $k$NN between each training set object and all objects in the other video clips provided in the training set. We can now define $\forall f \in \{velocity, pose, deep\} : \mu_f = \max_o\{s_f(\phi_f(o))\}$, and $\nu_f = \min_o\{s_f(\phi_f(o))\}$.

## 4.5 INFERENCE

Each inference clip $c = \{f_1, ..., f_n\}$ is fed frame by frame into both the optical flow estimator and the object detector. We then extract our attributed features from each object $o$. We compute an anomaly score for each attributed feature of each object $o$. The score for every frame is simply the maximum score across all objects. The final anomaly score is the sum of the individual feature scores normalized by our calibration parameters:

$$t(f) = \max_k\{\frac{s(\phi_{velocity}(o_k)) - \nu_{velocity}}{\mu_{velocity} - \nu_{velocity}}\} + \max_k\{\frac{s(\phi_{pose}(o_k)) - \nu_{pose}}{\mu_{pose} - \nu_{pose}}\}$$
$$+ \max_k\{\frac{s(\phi_{deep}(o_k)) - \nu_{deep}}{\mu_{deep} - \nu_{deep}}\} \quad (2)$$

We denote the anomaly score for every frame in a clip $c$ as $t(c) = \{t(f_1), ..., t(f_n)\}$. As we expect events to be prolonged, we smooth the results by applying a temporal 1-D Gaussian filter over $t(c)$.

## 5 EXPERIMENTS

### 5.1 DATASETS

Our experiments were conducted using three publicly available VAD datasets. Training and test sets are defined for each dataset, and anomalous events are only included during testing.

**UCSD Ped2.** The Ped2 dataset (Mahadevan et al., 2010) contains 16 normal training videos and 12 test videos at a $240 \times 360$ pixel resolution. Videos are gathered from a fixed scene with a camera above the scene and pointed downward. The training video clips contain only normal behavior of pedestrians walking, while examples of abnormal events are bikers, skateboarding, and cars.

**CUHK Avenue.** The Avenue dataset (Lu et al., 2013) contains 16 normal training videos and 21 test videos at $360 \times 640$ pixel resolution. Videos are gathered from a fixed scene using a ground-level camera. The training video clips contain only normal behavior. Examples of abnormal events are strange activities (e.g. throwing objects, loitering, and running), movement in the wrong direction, and abnormal objects.

**ShanghaiTech Campus.** The ShanghaiTech dataset (Liu et al., 2018) is the largest publicly available dataset for VAD. There are 330 training videos and 107 test videos from 13 different scenes at $480 \times 856$ pixel resolution. ShanghaiTech contains video clips with complex light conditions and camera angles, making this dataset more challenging than the other two. Anomalies include robberies, jumping, fights, car invasions, and bike riding in pedestrian areas.

### 5.2 IMPLEMENTATION DETAILS

We use ResNet50 Mask-RCNN (He et al., 2017) pretrained on MS-COCO (Lin et al., 2014) to extract object bounding boxes. To filter out low confidence objects, we follow the same configurations as in (Georgescu et al., 2021a). Specifically for Ped2, Avenue, and ShanghaiTech, we set confidence thresholds of 0.5, 0.8, and 0.8. In order to generate optical flow maps, we use FlowNet2 (Ilg et al., 2017). For our landmark detection, we use AlphaPose (Fang et al., 2017) pretrained on MS-COCO with $d = 17$ keypoints. We use a pretrained ViT B-16 (Dosovitskiy et al., 2020) CLIP (Radford et al., 2021) image encoder as our deep feature extractor. Our method is built around the extracted objects and flow maps. We use $H_{velocity} \times W_{velocity} = 224 \times 224$ to rescale flow maps. As for $H_{pose} \times W_{pose}$ rescaling, we calculate the average height and width from the bounding boxes of the train set and use those values. The lower resolution of Ped2 prevents objects from filling a histogram, and to extract pose representations, therefore we use $B = 1$ orientations and rely solely on velocity and deep representations. We use $B = 8$ orientations for Avenue and ShanghaiTech. When testing, for anomaly scoring we use $k$NN for the pose and deep representations with $k = 1$ nearest neighbors. For velocity, we use GMM with $n = 5$ Gaussians. Finally, the anomaly score of a frame represents the maximum score among all the objects within that frame.

Table 1: Frame-level AUROC (%) comparison. The best and second-best results are bolded and underlined, respectively.

| Year | Method | Ped2 | | Avenue | | ShanghaiTech | |
|---|---|---|---|---|---|---|---|
| | | Micro | Macro | Micro | Macro | Micro | Macro |
| ≤ 2019 | (Chaudhry et al., 2009) | 61.1 | - | - | - | - | - |
| | HOFM (Colque et al., 2016) | 89.9 | - | - | - | - | - |
| | S-RNN (Luo et al., 2017a) | 92.2 | - | 81.7 | - | 68.0 | - |
| | STAN (Lee et al., 2018) | 96.5 | - | 87.2 | - | - | - |
| | Frame-P (Liu et al., 2018) | 95.4 | - | 85.1 | - | 72.8 | - |
| | Mem-AE. (Gong et al., 2019) | 94.1 | - | 83.3 | - | 71.2 | - |
| | Ionescu et al. (2019) | 94.3 | 97.8 | 87.4 | 90.4 | 78.7 | 84.9 |
| | BMAN (Lee et al., 2019) | 96.6 | - | 90.0 | - | 76.2 | - |
| 2020 | Park et al. (2020) | 97.0 | - | 88.5 | - | 70.5 | - |
| | CAC (Wang et al., 2020) | - | - | 87.0 | - | 79.3 | - |
| | Scene-Aw (Sun et al., 2020) | - | - | 89.6 | - | 74.7 | - |
| | VEC (Yu et al., 2020) | 97.3 | - | 90.2 | - | 74.8 | - |
| | C-AE (Chang et al., 2020) | 96.5 | - | 86.0 | - | 73.3 | - |
| 2021 | AMMCN (Cai et al., 2021) | 96.6 | - | 86.6 | - | 73.7 | - |
| | Georgescu et al. (2021a) | 97.5 | 99.8 | 91.5 | 91.9 | 82.4 | 89.3 |
| | MPN (Lv et al., 2021) | 96.9 | - | 89.5 | - | 73.8 | - |
| | HF$^2$ (Liu et al., 2021a) | **99.3** | - | 91.1 | 93.5 | 76.2 | - |
| | Feng et al. (2021a) | 97.2 | - | 85.9 | - | 77.7 | - |
| | Georgescu et al. (2021b) | 98.7 | 99.7 | 92.3 | 90.4 | 82.7 | 89.3 |
| 2022 | (Ristea et al., 2022) | - | - | 92.9 | 91.9 | 83.6 | 89.5 |
| | DL-AC (Yang et al., 2022) | 97.6 | - | 89.9 | - | 74.7 | - |
| | JP (Wang et al., 2022) | 99.0 | **99.9** | 92.2 | 93.0 | 84.3 | **89.8** |
| 2023 | Yang et al. (2023) | 98.1 | - | 89.9 | - | 73.8 | - |
| | EVAL (Singh et al., 2023) | - | - | 86.0 | - | 76.6 | - |
| | Cao et al. (2023) | - | - | 86.8 | - | 79.2 | - |
| | FPDM (Yan et al., 2023) | - | - | 90.1 | - | 78.6 | - |
| | LMPT (Shi et al., 2023) | 97.6 | - | 90.9 | - | 78.8 | - |
| | Ours | 99.1 | **99.9** | **93.6** | **96.3** | **85.9** | 89.6 |

## 5.3 EVALUATION METRICS

Our study follows the popular evaluation metric in video anomaly detection literature by varying the threshold over the anomaly scores to measure the frame-level Area Under the Receiver Operation Characteristic (AUROC) with respect to the ground-truth annotations. We report two types of AU-ROC: (i) Micro-averaged AUROC, which is calculated by concatenating frames from all videos and then computing the score. (ii) Macro-averaged, which is calculated by averaging the frame-level AUROCs for each video. In most existing studies, micro-averaged AUROC is reported, while only a few report macro-averaged AUROC.

## 5.4 EXPERIMENTAL RESULTS

We compare our method and state-of-the-art from recent years in Tab. 1. The performance numbers of the baseline methods were directly taken from their original papers. We report both micro and macro average AUROC (when available) for the three publicly available most commonly used datasets: UCSD Ped2, CUHK Avenue, and ShanghaiTech.

**Ped2 Results.** Ped2 is a long-standing video anomaly detection dataset and has therefore been reported by many previous papers. Most methods obtained over 94% on Ped2, indicating that of the three public datasets, it is the simplest. While our method is comparable to the current state-of-the-art method (HF$^2$ Liu et al. (2021b)) in terms of performance, it also provides an interpretable representation. The near-perfect results of our method on Ped2 indicate it is practically solved.

**Avenue Results.** It is evident from previous works that Avenue is of a different complexity level than Ped2. Nevertheless, our method applied to this dataset obtained a new state-of-the-art AUROC

Table 2: Ablation study, frame-level AUROC (%) comparison. The best and second-best results are bolded and underlined, respectively.

| Pose Features | Deep Features | Velocity Features | Avenue | | ShanghaiTech | |
|:---:|:---:|:---:|:---:|:---:|:---:|:---:|
| | | | Micro | Macro | Micro | Macro |
| ✓ | | | 73.8 | 76.2 | 74.5 | 81.0 |
| | ✓ | | 85.4 | 87.7 | 72.5 | 82.5 |
| | | ✓ | 86.0 | 89.6 | 84.4 | 84.8 |
| ✓ | ✓ | | 89.3 | 88.8 | 76.7 | 84.9 |
| | ✓ | ✓ | _93.0_ | _95.5_ | 84.5 | 88.7 |
| ✓ | | ✓ | 86.8 | 93.0 | **85.9** | _88.8_ |
| ✓ | ✓ | ✓ | **93.6** | **96.3** | _85.1_ | **89.6** |

of 93.6% in terms of micro-averaged AUROC. Additionally, our method performance exceeds the current state-of-the-art by a considerable margin of 2.8%, reaching 96.3% macro-averaged AUROC.

**ShanghaiTech Results.** Our method outperforms all previous methods on the hardest dataset, ShanghaiTech, by a considerable margin. Accordingly, our method achieves 85.9% AUROC, while the highest performance previous methods have achieved is 84.3% (Jigsaw-Puzzle Wang et al. (2022)), surpassing the current state-of-the-art by a margin of 1.6%.

To summarize, our method achieves state-of-the-art performance on the three most commonly used public benchmarks. It outperforms all previous approaches without any optimization while utilizing representations that can be interpreted by humans.

## 5.5 ABLATION STUDY

We conducted an ablation study on Avenue and ShanghaiTech datasets to better understand the factors contributing to the performance of our method. We report anomaly detection performance of all feature combinations in Tab. 2. Our findings reveal that the velocity features provide the highest frame-level AUROC on both Avenue and ShanghaiTech, with 86.0% and 84.4% micro-averaged AUROC, respectively. In ShanghaiTech, our velocity features on their own are already state-of-the-art compared with all previous VAD methods. We expect this to be due to the large number of anomalies associated with speed and motion, such as running people and fast-moving objects, e.g. cars and bikes. The combination of velocity and pose results in an 85.9% AUROC in ShanghaiTech. The pose features are designed to detect unusual behavior, such as fighting between people and unnatural poses, as illustrated in Fig. 1 and App. A.2. However, we observe a slight degradation when we combine our attribute-based representation with the deep residual representation; this may be because deep representations bundle together many different attributes, and they are often dominated by irrelevant nuisance attributes that do not distinguish between normal and anomalous objects. As for Avenue, our attribute-based representation performs well when combined with the deep residual representation, resulting in state-of-the-art results of 93.6% micro-averaged AUROC and 96.3% macro-averaged AUROC. Overall, we have observed that using all three features was key to achieving state-of-the-art results.

## 5.6 FURTHER ANALYSIS & DISCUSSION

**Interpretable decisions.** We use a semantic attribute-based representation, which allows interpretation of the rationale behind decisions. This is based on the fact that our method categorizes frames as anomalous if their velocity and/or pose take an unusual value. The user can observe which attribute had an unusual value, this would indicate that the frame is anomalous in this attribute. To demonstrate the interpretability of our method, we present in Fig. 1 a visualization of most normal and anomalous frames in Avenue and ShangahiTech for each representation. High anomaly scores from the velocity representation are attributed to fast-moving (often non-human) objects. As can also be seen from the pose representation, the most anomalous frames contain anomalous human poses that are indicative of unusual behavior. Finally, our implicit deep representation captures concepts that cannot be adequately represented by our semantic attribute representation (for example, unusual objects). This complements the semantic attributes, obtaining the best of both worlds.

Table 3: Our final results when $k$NN is replaced by $k$-means. Frame-level AUROC (%) comparison.

| $k =$ | Avenue | | ShanghaiTech | |
|---|---|---|---|---|
| | Micro | Macro | Micro | Macro |
| 1 | 91.8 | 94.0 | 84.2 | 87.2 |
| 5 | 92.0 | 94.2 | 84.3 | 88.1 |
| 10 | 92.1 | 94.5 | 84.6 | 88.1 |
| 100 | 92.9 | 95.2 | 84.8 | 88.6 |
| All | **93.6** | **96.3** | **85.1** | **89.6** |

**Pose features for non-human objects.** We extract pose representations exclusively for human objects and not for non-human objects. We calculate the pose anomaly score for each frame by taking the score of the object with the most anomalous pose. Non-human objects are given a pose anomaly score of $-\infty$ and therefore do not contribute to the frame-wise pose anomaly score.

$k$**-Means as a faster alternative to** $k$**NN.** We can speed up $k$NN by reducing the number of samples via $k$-means. In Tab. 3, we compare the performance of our method when combined with velocity, pose, and deep features as well as its approximations based on $k$-means. Our method still uses $k$NN as the anomaly scores are calculated using the sum of distances to nearest neighbor means. This is much faster than the original $k$NN as there are fewer means than the number of objects in the training set. As can be seen, inference time can be significantly improved with a small loss in accuracy.

**What are the benefits of pretrained features?** Previous image anomaly detection work (Reiss et al., 2021) demonstrated that using feature extractors pretrained on external, generic datasets (e.g. ResNet on ImageNet classification) achieves high anomaly detection performance. This was demonstrated on a large variety of datasets across sizes, domains, resolutions, and symmetries. These representations achieved state-of-the-art performance on distant domains, such as aerial, microscopy, and industrial images. As the anomalies in these datasets typically had nothing to do with velocity or human pose, it is clear the pretrained features model many attributes beyond velocity and pose. Consequently, by combining our attribute-based representations with CLIP's image encoder, we are able to emphasize both explicit attributes (velocity and pose) derived from real-world priors and attributes that cannot be described by them, allowing us to achieve the best of both worlds.

**Why do we use an image encoder instead of a video encoder?** Newer and better self-supervised learning methods e.g. TimeSformer (Bertasius et al., 2021), VideoMAE (Tong et al., 2022), X-CLIP (Ni et al., 2022) and CoCa (Yu et al., 2022) are constantly improving the performance of pretrained video encoders on downstream supervised tasks such as Kinetics-400 (Kay et al., 2017). Hence, it is natural to expect that video encoders that utilize both temporal and spatial information will provide a higher level of performance than image encoders that do not. Unfortunately, in preliminary experiments, we found that features extracted by pretrained video encoders did not work as well a pretrained image features on the type of benchmark videos used in VAD. this result underscores the strong generalizability properties of pretrained image encoders, previously highlighted in the context of image anomaly detection. Improving the generalizability of pretrained video features in the one-class classification VAD setting is a promising avenue for future work.

## 6 CONCLUSION

Our paper proposes a simple yet highly effective attribute-based method that pushes the boundaries of video anomaly detection accuracy and interpretability. In every frame, we represent each object using velocity and pose representations, which is followed by density-based anomaly scoring. These simple velocity and pose representations allow us to achieve state-of-the-art in ShanghaiTech, the most complex video anomaly dataset. When we combine interpretable attribute-based representations with implicit deep representations, we achieve top video anomaly detection performance with a 99.1%, 93.6%, and 85.9% AUROC on Ped2, Avenue, and ShanghaiTech, respectively. We also demonstrated the advantages of our three feature representations in a comprehensive ablation study. Our method is highly accurate, interpretable, and easy to implement.

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

# A    APPENDIX

In the supplementary, we provide additional examples of frame-level scores predicted by our interpretable method as well as examples of localization. Furthermore, we provide information regarding the running time of our method.

## A.1    RUNNING TIME

We carried out all our experiments on a NVIDIA RTX 2080 GPU. Our preprocessing stage, which includes object detection and optical flow extraction, takes approximately 80 milliseconds (ms) per frame. It takes my method approximately 5 ms to compute the velocity extraction, pose extraction, and deep features extraction stages, combined with anomaly scoring. Our method runs at 12FPS with an average of 5 objects per frame.

## A.2    QUALITATIVE RESULTS

We provide visualization of the anomaly detection process for Avenue and ShanghaiTech in Fig. 4 and Fig. 5, where the anomaly curve shows the anomaly scores across all frames of a video. Our anomaly scores are highly correlated with the ground-truth occurrence of anomalous events. This demonstrates the effectiveness of our method. In Ped2, Fig. 6 and Fig. 7 demonstrate the effectiveness of our method, which can easily detect fast-moving objects such as trucks and bicycles. Accordingly, we can conclude that Ped2 has been practically solved based on the near-perfect results obtained by our method (as well as many others). Fig. 8 shows that our method is capable of detecting anomalies within a short timeframe. Fig. 9 and Fig. 10 provide more qualitative information regarding our method's ability to detect anomalies of various types. In this way, our method achieves a new state-of-the-art in Avenue and ShanghaiTech, surpassing other approaches by a wide margin.

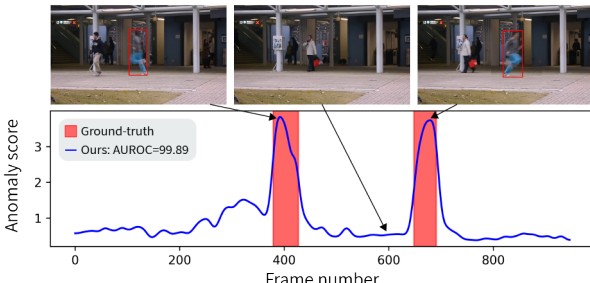

Figure 4: Frame-level scores and anomaly localization examples for test video 04 from Avenue. Best viewed in color.

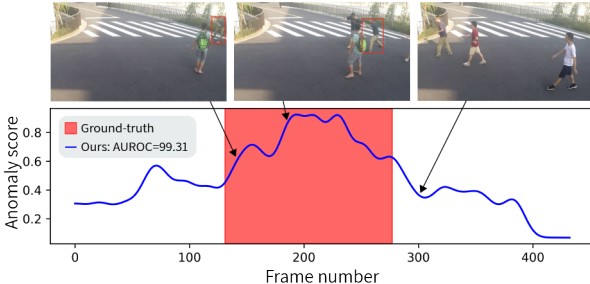

Figure 5: Frame-level scores and anomaly localization examples for test video 03_0059 from ShanghaiTech. Best viewed in color

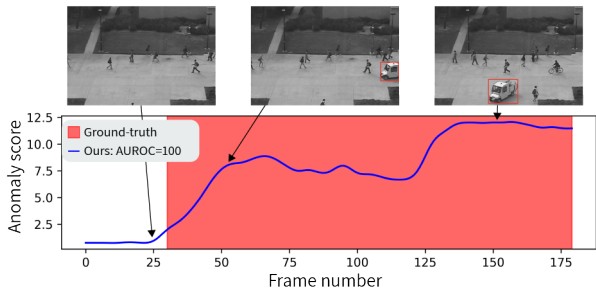

Figure 6: Frame-level scores and anomaly localization examples for test video 04 from Ped2. Best viewed in color.

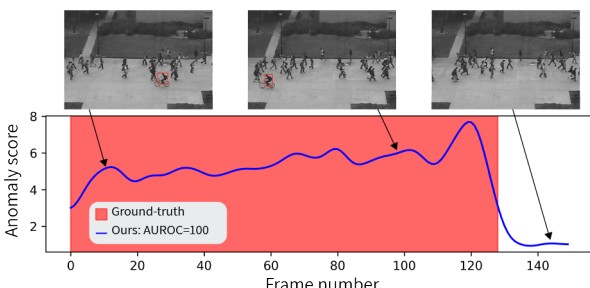

Figure 7: Frame-level scores and anomaly localization examples for test video 05 from Ped2. Best viewed in color.

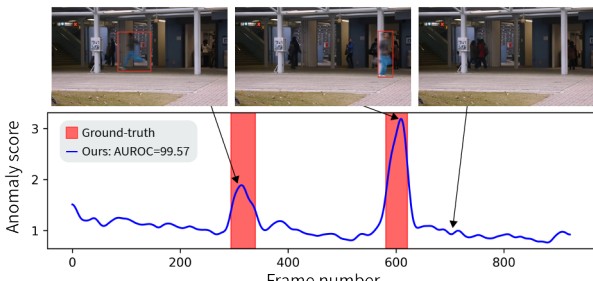

Figure 8: Frame-level scores and anomaly localization examples for test video 03 from Avenue. Best viewed in color.

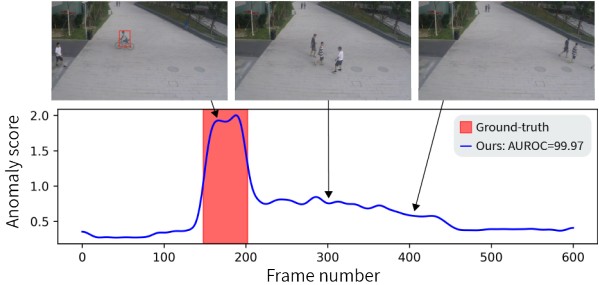

Figure 9: Frame-level scores and anomaly localization examples for test video 01_0025 from ShanghaiTech. Best viewed in color.

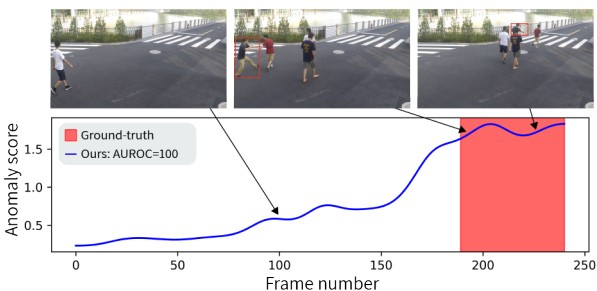

Figure 10: Frame-level scores and anomaly localization examples for test video 07_0048 from ShanghaiTech. Best viewed in color.

