# OpenReview forum: "Video Anomaly Detection via Semantic Attributes"
_ICLR.cc/2024/Conference — ICLR 2024 Conference Withdrawn Submission_

### Official Review · Reviewer_p8Zy · 2023-10-16

**Soundness:** 3 good
**Presentation:** 4 excellent
**Contribution:** 1 poor
**Rating:** 5
**Confidence:** 5

**Summary:**

The paper introduces an anomaly detection method for the video domain, which combines velocity (via FlowNet), pose (via AlphaPose) and deep features (via CLIP) extracted from objects detected by a pre-trained object detector (Mask R-CNN). The authors perform experiments on three video databases (Avenue, ShanghaiTch, Ped2) to compare the proposed method with several state-of-the-art methods.

**Strengths:**

- The proposed method obtains very good results.
- The paper is easy to follow.

**Weaknesses:**

- The proposed method seems to be a simple combination of existing (even pre-trained) components. Therefore, in my opinion, the technical novelty is very limited.
- The pose features can only be extracted for humans. It is not clear what happens if the object is not a human.
- The authors should relate to existing interpretable video anomaly detection methods, e.g. [1, 2].
- "It is surprising that our simple velocity and pose representations achieves state-of-the-art performance on the largest and most complex VAD dataset, with 85.9% AUROC in ShanghaiTech" => This statement is incorrect. ShanghaiTech is definitely not the largest database, e.g. XD-Violence or UCF-Crime are much larger. Moreover, the simple approach does not seem to work all by itself (it still needs to be coupled with deep features). In the end, it would be interesting to see if the simple approach proposed by the authors would work on these truly large-scale datasets.
- On Avenue, there are some methods obtaining better results (see [3]). These should be added to Table 1.
- Some recent works used RBDC/TBDC to estimate localization performance. It would be interesting to see how the proposed method is evaluated against RBDC/TBDC.
- The proposed method seems to use may heavyweight systems. However, the processing for video anomaly detection is expected to happen in real-time at the standard video FPS rate, e.g. 25/30 FPS. The method seems to be twice as heavier.
- The interpretability of the method seems rather shallow, i.e. it only indicates if the anomaly is related to pose, velocity or deep features. If deep features are the cause, we are back to square one.
- Typo fixes:
  - "VAD. this result underscores" should be "VAD. This result underscores".
  - "It takes my method" should be "It takes our method"

[1] Doshi, Keval, and Yasin Yilmaz. "Towards interpretable video anomaly detection." In Proceedings of the IEEE/CVF Winter Conference on Applications of Computer Vision, pp. 2655-2664. 2023.

[2] Wang, Yizhou, Dongliang Guo, Sheng Li, and Yun Fu. "Towards Explainable Visual Anomaly Detection." arXiv preprint arXiv:2302.06670 (2023).

[3] https://paperswithcode.com/sota/anomaly-detection-on-chuk-avenue

**Questions:**

- How does the time of the proposed method, which is based on k-NN, scale with the size of the training data? It would be nice to see a graph.

---

### Official Review · Reviewer_W9ta · 2023-10-30

**Soundness:** 1 poor
**Presentation:** 2 fair
**Contribution:** 1 poor
**Rating:** 3
**Confidence:** 5

**Summary:**

This paper presents a semantic attribute-based approach for improving the accuracy and interpretability of video anomaly detection. The method represents each object in each video frame in terms of its speed and pose, and uses a density-based approach to compute the anomaly score. The method achieves state-of-the-art performance on three publicly available video anomaly detection datasets, while also providing information on why anomalies occur.

**Strengths:**

1) The paper introduces a simple but effective attribute-based method for video anomaly detection. The use of attribute-based representations allows for interpretability and understanding of the reasoning behind the system's decisions.
2) The proposed method achieves state-of-the-art performance on three commonly used video anomaly detection datasets, Ped2, Avenue, and ShanghaiTech. This demonstrates the effectiveness of the method in accurately detecting anomalies.

**Weaknesses:**

1)The method does not make full use of the temporal information in the video, but only extracts features based on a single frame or two adjacent frames. This may result in ignoring anomalous events that need to take into account long-term dynamic changes, such as the sudden disappearance or appearance of people. The authors should consider using more frames to construct a feature representation of the object, or using some model that can capture timing dependencies, such as recurrent neural networks or transformers.
2) The method uses a pre-trained CLIP encoder as a deep feature extractor, but no specific experimental details or rationale are given. The authors should explain why CLIP was chosen over other image encoders and how the CLIP encoder was fine-tuned or adapted for the video anomaly detection task. In addition, the authors should have given the parameter settings and performance comparison of the CLIP encoder on each dataset.
3)The method uses a fixed threshold to determine whether each frame is anomalous or not, but does not explain how this threshold is determined. The authors should give the method of threshold selection and sensitivity analyses, as well as the threshold values taken on different datasets. In addition, the authors should explore the possibility of using adaptive thresholds to accommodate variations in different scenes and objects.
4)There are some references in the article that are not given in a standardised format. For example, in the last sentence of the third paragraph on the second page, "In practice, we used AlphaPose (Fang et al., 2017), which we found to work well.", the specific citation of AlphaPose should be given; in the first sentence of the first paragraph on page 4.

**Questions:**

1)Can you explain why you chose the CLIP encoder?
2)How the CLIP encoder was fine-tuned or adapted on the video anomaly detection task?
3) Can you give the selection criteria and sensitivity analyses for giving thresholds?
4)How did you determine the contribution of each feature to the final anomaly score t? Did they all contribute equally?
5)Can you give the parameter settings and performance comparison of CLIP encoder on various datasets?

---

### Official Review · Reviewer_EZHR · 2023-10-31

**Soundness:** 2 fair
**Presentation:** 3 good
**Contribution:** 2 fair
**Rating:** 3
**Confidence:** 4

**Summary:**

The paper introduces a simple yet powerful method for Video Anomaly Detection (VAD) using attribute-based representations. Objects in video frames are represented by their velocity and pose, and anomalies are detected using a density-based approach. This straightforward technique achieves state-of-the-art performance on challenging datasets like ShanghaiTech, with AUROC scores of 99.1%, 93.6%, and 85.9% on Ped2, Avenue, and ShanghaiTech respectively.

**Strengths:**

1. Originality: Introduces a novel, simple attribute-based approach to Video Anomaly Detection (VAD), focusing on velocity and pose.

2. Quality: Achieves state-of-the-art results on challenging datasets, backed by a comprehensive ablation study.

3. Clarity: Presents ideas and results in a clear, accessible manner.

**Weaknesses:**

1. Dataset Specificity:
The proposed method seems tailored specifically to the peculiarities of the three test datasets (Ped2, Avenue, and ShanghaiTech). It effectively distinguishes between certain object types like bicyclists and pedestrians but may not be as effective in more varied scenarios.

2. Limited Attribute Scope:
While the addition of velocity and pose attributes improves performance on these specific datasets, it's questionable how this approach would fare in more complex and diverse datasets, such as UCFCrime.

3. Lack of Generalization:
The method demonstrates limited generalizability beyond the tested scenarios. Its effectiveness in other types of anomaly detection tasks, particularly in industrial settings or datasets with different anomaly types, remains unproven.
4. Narrow Inspirational Scope:
Due to its dataset-specific nature and limited attribute scope, the method offers minimal inspiration or methodological guidance for broader anomaly detection challenges. The findings might not be readily applicable or extendable to other contexts or more intricate anomaly detection tasks.

**Questions:**

See the weaknesses.

---

### Official Review · Reviewer_fxdo · 2023-11-01

**Soundness:** 2 fair
**Presentation:** 3 good
**Contribution:** 2 fair
**Rating:** 5
**Confidence:** 4

**Summary:**

This paper proposes a video anomaly detection method that achieves state-of-the-art performance while also being interpretable. The key contributions are:

- The method represents each object using two semantic attributes: velocity and human pose. These allow automatic interpretation of anomaly decisions based on whether the velocity or pose are unusual.

- Despite the simplicity of the representations, the method achieves SOTA results on the three most commonly used VAD datasets: Ped2, Avenue, and ShanghaiTech. It surpasses prior work substantially on ShanghaiTech, the largest and most complex dataset.

- The velocity and pose representations are complemented with an implicit deep representation to model residual attributes not captured by velocity/pose. This achieves the best of both interpretable attributes and powerful deep features.

- The method is easy to implement, requiring only off-the-shelf components like object detectors and optical flow.

- Comprehensive experiments demonstrate the effectiveness of the approach, including ablation studies and visualization of anomaly decisions.

**Strengths:**

**Originality:** The idea of using semantic attributes like velocity and pose as representations for anomaly detection is novel and has not been explored before. Prior work relied on less interpretable features like reconstruction errors or deep embeddings. Representing objects by velocity and pose provides inherent interpretability.

**Quality:** The method achieves impressive results, outperforming prior art substantially on the largest VAD dataset, ShanghaiTech. The ablation studies provide insight into the relative importance of the different components. The experiments are comprehensive and compare favorably to a large number of prior approaches.

**Clarity:** The paper is well-written and easy to follow. The methodology is clearly explained, with helpful diagrams illustrating the components. The results are presented in a clear format with tables comparing to prior work.

**Significance:** The ability to provide interpretable rationales for anomaly decisions is critical for deploying VAD in real-world applications like surveillance. Opaque neural networks can make mistakes that humans cannot understand. This work shows that interpretable semantic attributes can be surprisingly effective for VAD. The significance goes beyond pushing SOTA - it demonstrates that interpretability does not require sacrificing accuracy.

**Weaknesses:**

- More analysis could be provided on failure cases to understand limitations of the approach. Where does the method break down? What types of anomalies are missed?

- The motivations for some design choices are unclear. For example, why are velocity and pose specifically chosen as the semantic attributes? How were hyperparameters like number of velocity bins selected?

- The method relies on off-the-shelf components like optical flow and object detection. How robust is it to errors in these modules? Could end-to-end training help?

- More analysis could be provided understanding what concepts are captured by the deep representation versus the attributes. Are they complementary?

**Questions:**

Please see the weaknesses above.

---

### Author Response · Authors · 2023-11-17

**Motivations for design choices:** The choice of velocity and pose as semantic attributes stems from their established significance in prior works, specifically in the context of video anomaly detection (Markovitz et al., 2020; Georgescu et al., 2021a). They were shown to be very effective on the tested datasets. There are clearly some anomalies that they cannot capture, which motivates including other features in the detector. Such attributes can be specified by the user or discovered through a review of the existing literature.

**Understanding concepts captured by deep vs. semantic attributes:** Our findings indicate that the primary performance gains are attributed to the velocity and pose features, while deep features serve as a complementary, residual component. This nuanced understanding will be conveyed more explicitly, enhancing the clarity of the proposed approach.

**Can end-to-end training help?** This might help, although as results are already better than state-of-the-art we leave this direction to future work.

**Datasets and attribute scope:**  We selected velocity and pose as the semantic attributes in our method due to their prominence in prior VAD works.  They were shown to be very effective on the tested datasets. There are clearly some anomalies that they cannot capture, which motivates including other features in the detector. Such attributes can be specified by the user or discovered through a review of the existing literature.

**Inspirational scope:** An open question that arises from our work is the design of practical attribute guidance methods for learning representations in real-world anomaly detection scenarios. Future research should explore methods that leverage semantic attributes, taking into account factors such as prior knowledge availability and reliability. Addressing the challenge of providing guidance in real-world scenarios, where anomalies may manifest in diverse and unexpected ways, will be crucial to advancing the field. Our work serves as a stepping stone, emphasizing the need for future research incorporating diverse semantic attributes. By doing so, the field can advance towards more effective, interpretable, and adaptable anomaly detection systems tailored to real-world complexities.

**Temporal information:** The demonstration that state-of-the-art performance can be achieved using only two adjacent frames highlights a potential deficiency in current video anomaly detection datasets. We also believe that this emphasizes the need for datasets that better capture the temporal nuances of anomalous events.

**CLIP encoder:** We used the CLIP image encoder as an off-the-shelf component without any fine-tuning, and this information is in the current manuscript. We did include a comprehensive ablation study in the manuscript, reporting the dataset breakdown results for the CLIP encoder, thus addressing the reviewer's concern about parameter settings and performance comparison.

**Thresholds:** We appreciate your suggestion regarding the threshold determination method. As we use the AUROC metric, which is standard in the video anomaly detection literature, threshold selection becomes less critical.